# Does the Severity of Obstructive Sleep Apnea Have an Independent Impact on Systemic Inflammation?

**DOI:** 10.3390/medicina57030292

**Published:** 2021-03-22

**Authors:** Romana Suša, Vojislav Ćupurdija, Ljiljana Novković, Miloš Ratinac, Slobodan Janković, Danijela Đoković, Jovan Jovanović, Katarina Pantić, Stefan Simović, Danijela Bazić-Sretenović, Ivan Čekerevac

**Affiliations:** 1Department of Internal Medicine, Faculty of Medical Sciences, University of Kragujevac, 34000 Kragujevac, Serbia; vojacup@gmail.com (V.Ć.); ljiljanan1@hotmail.com (L.N.); milos.ratinac@gmail.com (M.R.); jovanovmejl1@gmail.com (J.J.); katarinapantic@yahoo.com (K.P.); simovicst@gmail.com (S.S.); bazicdanijela@yahoo.com (D.B.-S.); icekerevac@gmail.com (I.Č.); 2Clinic for Pulmonology, University Clinical Center Kragujevac, 34000 Kragujevac, Serbia; 3Department of Pharmacology and Toxicology, Faculty of Medical Sciences, University of Kragujevac, 34000 Kragujevac, Serbia; slobnera@gmail.com; 4Department of Clinical Pharmacology, University Clinic Center Kragujevac, 34000 Kragujevac, Serbia; 5Psychiatry Clinic, University Clinical Center Kragujevac, 34000 Kragujevac, Serbia; danijela-kckragujevac@hotmail.com; 6Clinic for Cardiology, University Clinical Center Kragujevac, 34000 Kragujevac, Serbia; 7Clinic for Rheumatology, Clinical Immunology and Allergology, University Clinical Center Kragujevac, 34000 Kragujevac, Serbia

**Keywords:** obstructive sleep apnea, systemic inflammation, nutritional status, body composition

## Abstract

*Background and Objectives*: This paper aims to show whether obstructive sleep apnea (OSA) severity increases the level of systemic inflammation markers regardless of body mass index (BMI) and body composition. *Materials and Methods*: In total, 128 patients with OSA were included in the study. Examinees were divided into two groups: one with mild OSA (apnea–hypopnea index (AHI) < 15) and one with moderate and severe OSA (AHI ≥ 15). Nutritional status was assessed using body mass index, body composition by dual X-ray absorptiometry. Systemic inflammation was assessed on the basis of plasma concentrations of tumor necrosis factor-α (TNF-α), interleukin-6 (IL-6), and serum level of C-reactive protein (CRP). *Results*: We found elevated mean values of the evaluated systemic inflammation markers (CRP, TNF-α, IL-6) in a group with AHI ≥ 15, although there was no statistical significance. Our research found a significant positive correlation with BMI (*r* = 0.633, *p* < 0.001), as well as with body fat percentage (*r* = 0.450, *p* = 0.024) and serum CRP values. Significant correlation was found between the plasma IL-6 concentration and body fat percentage (FM%) (*r* = 0.579, *p* = 0.003) and lean body mass (*r* = −0.501, *p* = 0.013). Multivariate regression analysis did not show any independent predictor (parameters of OSA, nutritional status, body composition) of the systemic inflammation markers. *Conclusions*: Neither one tested parameter (nutritional status and body composition) of the severity of OSA was identified as an independent prognostic factor for the severity of systemic inflammation in patients with OSA.

## 1. Introduction

Obstructive sleep apnea (OSA) is a disorder that is characterized by recurrent episodes of interrupted breathing lasting more than 10 s (apnea), or by a reduction in airflow (hypopnea) during sleep associated with sleep fragmentation, awakenings, and decreased oxygen saturation. Since respiratory control is considered to be imperfect, up to five respiratory episodes during 60 min of sleep are considered normal [1]. There are several ways to show the severity of desaturation during sleep as a consequence of OSA. The oxygen desaturation index (ODI) is the most commonly used metric for evaluation and is characterized by the number of desaturations (saturation (SatO2) reduction ≥ 3%) per hour of sleep, the mean blood oxygen saturation, the minimal saturation, and the proportion of sleep hours expressed as a percentage with hemoglobin oxygen saturation below 90% (Sat < 90%). It is assumed that the cyclical occurrence of hypoxia-reoxygenation (intermittent hypoxia) that characterizes OSA is one of the prime mechanisms that generate the formation of free radicals, similar to ischemia-reperfusion-related injuries in coronary disease. It is believed that intermittent hypoxia leads to mitochondrial dysfunction, the activation of enzymes that use oxygen, such as xanthine oxidase or nicotinamide adenine dinucleotide phosphate (NADPH), the activation of leukocytes, and endothelial cell dysfunction, which in turn leads to the synthesis of free radicals [2]. Intermittent hypoxia can, as a consequence, have an increased expression of systemic inflammatory mediators [3]. Fragmented sleep and sleep deprivation induce an increase in cytokine levels, which may give way to the inflammatory response. Several studies show that patients with OSA have increased levels of systemic inflammatory mediators, including intercellular adhesion molecules (ICAM), coagulation factors (F VIII, tissue factor), and C-reactive proteins (CRPs) [4,5,6,7,8]. Elevated levels of tumor necrosis factor-α (TNF-α), interleukin-1β (IL-1β), and interleukin-6 (IL-6) have been found in patients with OSA [9,10]. On the other hand, data from medical publications show that aside from mechanical factors, local inflammation can contribute to the development of OSA as well. Evidence of this can be seen in the histological biopsy of the uvula in patients with OSA, which showed histopathological abnormalities, including subepithelial edema and inflammatory cell infiltration [3].

Adipose tissue, especially visceral fat, is where a vigorous synthesis of IL-6 occurs, which further leads to elevated levels of CRP in obese individuals. Visceral adipocytes synthesize several cytokines and are metabolically more active than subcutaneous fat. It is still not fully understood whether elevated inflammatory mediator levels in OSA are independent of obesity.

This paper aims to show whether OSA severity increases the level of systemic inflammation markers regardless of body mass index (BMI) and body composition.

## 2. Materials and Methods

The Ethics Committee approval to conduct this trial was requested and received on 19.06.2018, approval No 01/18-2488. Before the patients were included in this examination, the purpose of the examination and details of their participation were explained to them, and they were asked to sign the informed consent form. One copy of this document was given to the examinee/patient and the other was retained on site with study documentation.

The study included 128 patients with OSA. Patients with concomitant diseases, characterized by the existence of systemic inflammation, were excluded, except for those with obesity. All examinees underwent the sleep study nocturnal respiratory polygraphy using the Philips Alice PDx sleep recorder. The apnea–hypopnea index (AHI), desaturation index (ODI), minimum saturation (min SatO2), average saturation (mean SatO2), and the percentage of time with saturation below 90% (SatO2 < 90%) were determined. In line with the results of the sleep study based on the AHI, the examinees were divided into two groups: one with mild OSA (AHI < 15) and one with moderate and severe OSA (AHI ≥ 15). All examinees underwent nutritional status assessment using BMI = body mass (BM)/body height^2^ (kg/m^2^). Body compositions were determined by a dual X-ray absorptiometry (DXA) whole-body scan (Hologic QDR-4000). The absolute value of body fat mass (FM) in kg, body fat percentage (FM%), fat mass index (FMI = FFM/ body height^2^ (kg/m^2^)), lean body mass (FFM) in kg, lean body mass index (FFMI = FFM/body height^2^ (kg/m^2^)) and central FM (kg) were examined (Figure 1). Data from the National Health and Nutrition Examination Survey (NHANES) for FM% reference values using DXA measuring was used (13).

Systemic inflammation was assessed on the basis of the plasma concentrations of TNF-α, determined by the AviBion Human Leptin ELISA kit (Orgenium, Helsinki, Finland); plasma concentrations of IL-6, determined by the AviBion Human Leptin ELISA kit (Orgenium, Helsinki, Finland); and serum CRP concentrations (Olympus AU 400, immunoturbidimetric method for the quantitative determination of C-reactive protein).

The descriptive statistics method was used to identify the general characteristics of the groups of examinees, as well as the results obtained from the completed test. An independent t-test was used to compare the arithmetic mean of the measured characteristics of the two populations. The correlation was used to test the relationship between the parameters, and its presentation and the interpretation of the significance were performed using linear correlation. The influence of individual variables as independent predictors for the observed parameters was examined by using the multivariate regression analysis.

## 3. Results

Patients were divided into two groups according to AHI (cut-off = 15). There were 56 (43.8%) patients (44 males) in the group with AHI < 15 and 72 (52.2%) patients (48 males) in the group with AHI ≥ 15.

The mean values of the parameters for the sleep study, nutritional status assessment, body composition, and systemic inflammation in test groups (t-test) are presented in Table 1. We did not detect any statistical significance for the mean levels of the plasma TNF-α and IL-6 concentrations as markers of systemic inflammation between these two groups of examinees (Table 1).

We found a significant positive correlation between the BMI and body fat percentage (FM%) and the serum CRP values (Figure 2 and Figure 3).

A univariate regression analysis determined the following parameters as predictors of serum CRP values: min SatO2, BMI, FM%, and FMI (Table 2).

Using a multivariate regression analysis, none of the parameters for assessing the state of nutrition, body composition, and sleep study parameters were identified as predictors of serum CRP (Table 3).

Using univariate regression analysis, none of the parameters for assessing the state of nutrition, body composition, and sleep study parameters were identified as predictors of the plasma concentrations of IL-6 and TNF-α.

We found a significant positive association between serum IL-6 concentration and SatO2 of less than 90% (Figure 4), as well as an association between IL-6 and FM% (Figure 5). There was a significant negative association between IL-6 and lean body mass (FFM) (Figure 6).

## 4. Discussion

We analyzed systemic inflammation in groups with different severities of OSA. We found elevated mean values of the evaluated systemic inflammation markers (CRP, TNF-α, and IL-6), although there was no statistical significance in a group with AHI ≥ 15 in comparison to the group with AHI < 15. Our research found a significant positive correlation with BMI, as well as with body fat mass (FM%, FMI), and serum CRP values. Multiple regression analysis; however, did not show any independent predictor of the serum CRP value. A meta-analysis of Nadeem et al. analyzed 51 studies that evaluated the association of systemic inflammation markers and OSA (30 studies about CRP, 19 about TNF-α, and 18 about IL-6) [11,12]. The level of inflammatory markers was higher in the group with OSA in comparison to the control group. The standardized mean difference was 1.77 for CRP, 1.03 for TNF-α, and 2.16 for IL-6. This meta-analysis showed that patients with OSA had a significantly higher level of CRP in comparison to the control group. Age, together with BMI and AHI, was found to have an important influence on all inflammation markers. Guven et al. showed an elevated level of serum high sensitivity CRP (hs-CRP) in patients with OSA in comparison to the control group without OSA [13]. A significant positive correlation of hs-CRP with BMI and AHI was found, but multiple regression analysis showed that hs-CRP was only associated with AHI, irrespective of BMI. Lee et al. showed that in patients with OSA, the increase of hs-CRP positively correlated with BMI, AHI, and tonsil size. However, only AHI could predict significantly elevated hs-CRP after multiple linear regression analysis [14].

Various other studies did not show the association between AHI and CRP concentrations. They reported that obesity, rather than OSA or nocturnal hypoxemia, is the key predictor of the elevated level of CRP in patients with OSA. Barcelo et al. showed that CRP levels are significantly higher in obese patients with OSA when compared to non-obese patients with OSA [15]. In the study, Sharma et al. examined the level of hs-CRP in newly diagnosed patients with OSA [16]. These patients were divided into three groups: OSA (AHI > 5), obese without OSA (BMI > 25, AHI < 5), and non-obese without OSA (BMI < 25, AHI < 5). The highest mean value of hs-CRP was found in the group of obese without OSA. There was a significant positive correlation between hs-CRP and BMI, whereas there was none with AHI. Moreover, there was a significant association of hs-CRP with BMI and AHI, even after age adjustment, whereas there was no significant association of hs-CRP with the severity of OSA (based on AHI), even after BMI and age adjustment. Multiple regression analysis identified only BMI as significantly associated with the serum hs-CRP levels [16]. The Guilleminault et al. study showed that CRP levels in patients with OSA significantly correlated with BMI, and that BMI is the only value to correlate independently with an elevated level of CRP in patients with OSA [17]. Ryan et al. showed a similar level of CRP in male patients without OSA, patients with mild to moderately severe OSA, and patients with severe OSA. However, the level was markedly elevated in obese patients with severe OSA. It was concluded that the level of CRP does not correlate with the severity of OSA in male patients with OSA, but rather correlates independently with obesity [18].

We found elevated mean values of the serum TNF-α in the group with AHI ≥ 15, although there was no statistical significance. Several studies showed that patients with OSA have higher levels of TNF-α [8,19], with levels of TNF-α correlating significantly with the severity of OSA and nocturnal hypoxemia [20]. In contrast, Fornadi et al. did not find any significant correlation between concentrations of TNF-α and AHI [21]. A slightly elevated, but not negligible level of IL-6 was found in patients in this study. Our study supports the claim that in patients with OSA, a Th1 (T helper-1) type of cytokine response is activated. Analyzing the association of IL-6 concentrations in blood plasma with the parameters of the sleep study, we did not find any significant correlation with AHI. The only important positive correlation was between IL-6 and SatO2 < 90% (as a percentage of the time). Many cross-sectional studies showed a higher level of IL-6 in patients with OSA and a significant correlation between IL-6 and AHI [22,23]. Mehra et al. found that subjects with moderate to severe OSA had markedly higher morning and evening values of IL-6 when compared with subjects without OSA, but these differences were not statistically significant after adjustments for subject characteristics [24]. Inflammatory cytokine IL-6 is responsible for the synthesis of CRP in the liver. Likewise, IL-6 is produced in visceral adipose tissue in obese individuals. Evidence of this can be found in our study results, as we found a significant positive association between IL-6 concentration and fat mass (adipose tissue percentage). However, multiple regression analysis did not identify a single parameter as an independent predictor of IL-6 concentration. Our study found a significant negative association between IL-6 concentrations in blood plasma and lean body mass, which can bring attention to the role of systemic inflammation in body composition changes in patients with OSA. Kosacka et al. found a significant decrease in muscle mass in patients with OSA when compared to the control group [25]. The reason behind these discrepancies could be found in the methodology of previous studies and the small number of patients included.

A possible limitation of our study is the inclusion of patients who had already been diagnosed with cardiovascular or metabolic diseases.

## 5. Conclusions

OSA severity does not increase the level of systemic inflammation markers regardless of body mass index (BMI) and body composition. Neither the severity of OSA, the nutritional status, or the body composition were identified as independent prognostic factors for the severity of systemic inflammation in patients with OSA.

## Figures and Tables

**Figure 1 medicina-57-00292-f001:**
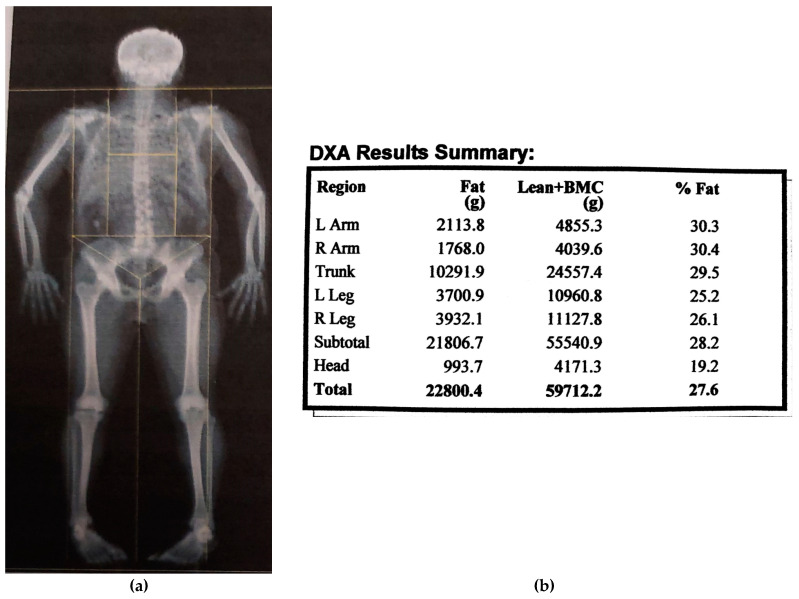
(**a**) Dual X-ray absorptiometry (DXA) scan and (**b**) DXA results summary.

**Figure 2 medicina-57-00292-f002:**
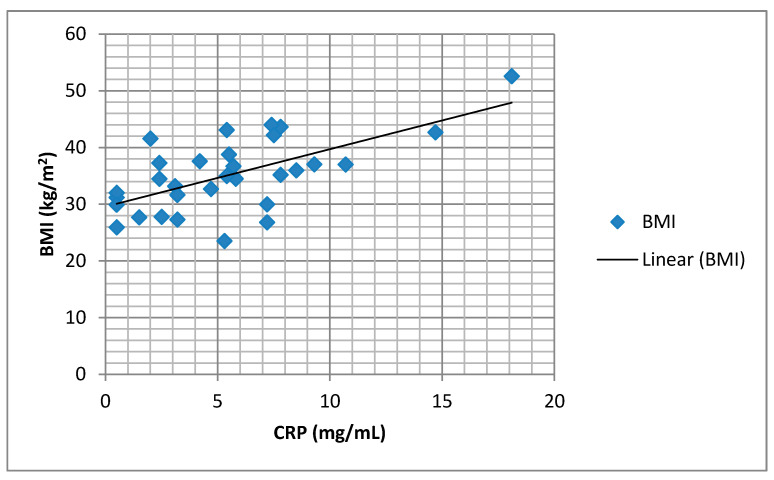
Correlation of serum CRP and body mass index (BMI) (*r* = 0.633, *p* = 0.000).

**Figure 3 medicina-57-00292-f003:**
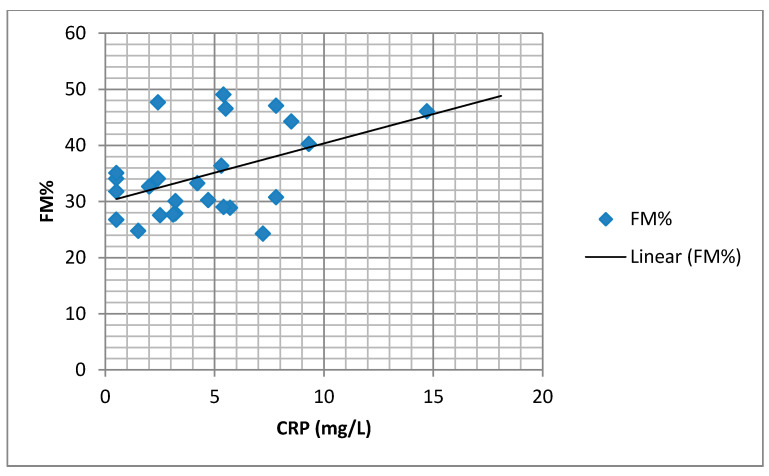
Correlation of serum CRP and body fat percentage (FM%) (*r* = 0.450, *p* = 0.024).

**Figure 4 medicina-57-00292-f004:**
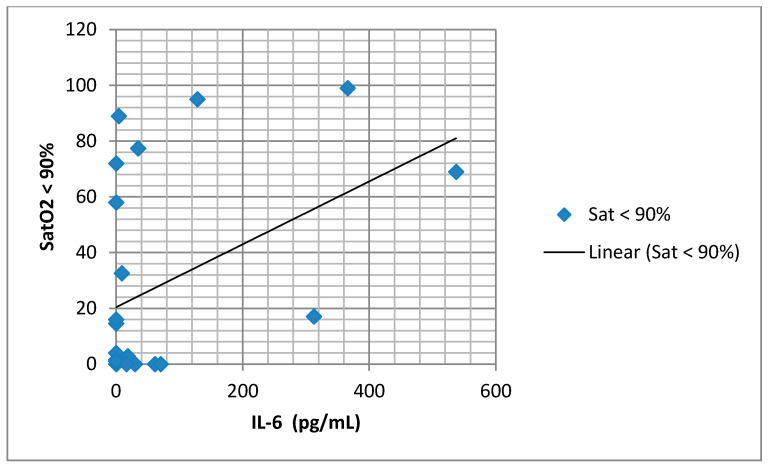
Correlation of serum interleukin-6 (IL-6) and saturation (SatO2) < 90% (*r* = 0.437, *p* = 0.033).

**Figure 5 medicina-57-00292-f005:**
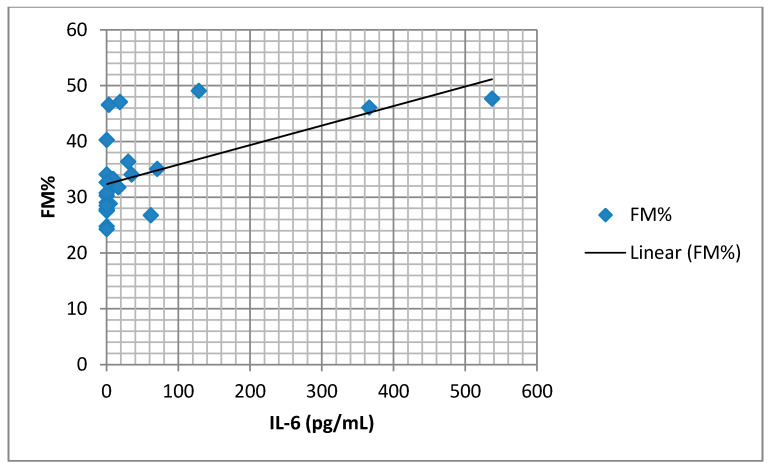
Correlation of serum IL-6 and FM% (r = 0.579, *p* = 0.003).

**Figure 6 medicina-57-00292-f006:**
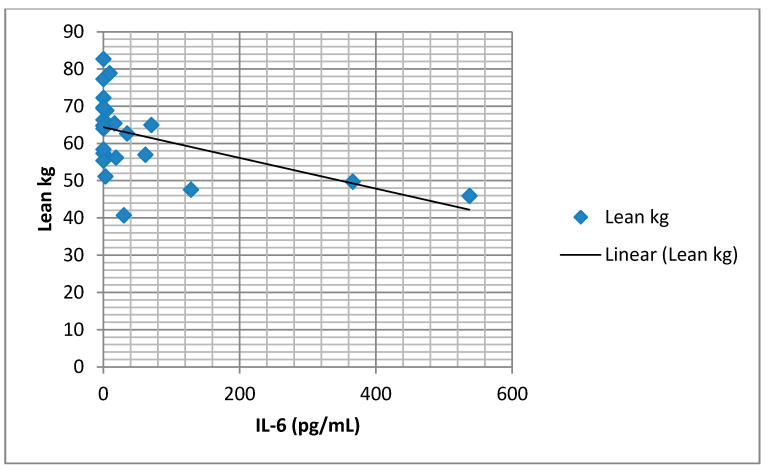
Correlation of serum IL-6 and lean body mass (FFM) (*r* = −0.501, *p* = 0.013).

**Table 1 medicina-57-00292-t001:** Demographic characteristics and mean values of obstructive sleep apnea (OSA), nutritional status assessment, body composition and systemic inflammation parameters in test groups.

Examined Parameters X ±SD (min-max)	AHI < 15	AHI ≥ 15	*p*
Age (years)	42.57 ± 13.63	52.44 ± 11.38	0.033
AHI n (min-max)	10.2 (5.1–14.8)	24.3 (15.1–82)	<0.001
ODI	4.63 ± 7.06 (0–23.5)	21.56 ± 14.45	<0.001
Mean SatO2 (%) (min/max)	93.54 ± 2.90 (88–97)	85.63 ± 12.52 (43–96)	0.035
Min SatO2 (%) (min/max)	86.08 ± 4.80 (80–94)	67.75 ± 13.26 (39–85)	<0.001
SatO2 < 90% (%) (min/max)	4.90 ± 10.04 (0–32.6)	41.86 ± 38.43 (1–99)	0.001
BMI (kg/m^2^)	32.29 ± 5.58 (23.5–42.2)	36.55 ± 6.74 (26.8–52.6)	0.156
FM%	31.97 ± 4.13 (26.8–40.3)	34.25 ± 9.36 (24.3–49.1)	0.234
FMI (kg)	31.68 ± 7.78	34.89 ± 9.64	0. 483
FFM (kg)	64.01 ± 12.14	59.0 ± 10.16	0.929
FFMI	20.80 ± 3.46	20.8 ± 4.12	0.172
CentrFM (kg)	16.84 ± 3.98 (11.28–23.11)	17.87 ± 3.84 (12.52–26.50)	0.51
TNF-α(pg/mL)	4.26 ± 3.92 (4–6–9.9)	10.32 ± 2.32	0.929
IL-6 (pg/mL)	38.41 ± 12.5 (0–333)	65.17 ± 21.90 (0–537)	0.964
CRP (mg/L)	4.52 ± 3.40(0.5–10.7)	6.2 ± 4.43 (0.5–18.1)	0.261

**Table 2 medicina-57-00292-t002:** Univariate regression analysis for serum C-reactive protein (CRP) values.

Risk Factors Examined	Univariate	
	^#^ B (95%CI)	*p*
Age	0.089 (−0.021–0.199)	0.109
Gender	2.646 (−0.536–5.828)	0.100
AHI	0.021 (−0.031–0.072)	0.419
ODI	0.047 (−0.008–0.101)	0.09
Mean SatO2	−0.076 (−0.239–0.086)	0.342
MinSatO2	−0.157 (−0.262–−0.053)	0.005 *
SatO2 < 90%	0.033 (−0.019–0.084)	0.202
BMI	0.396 (0.212–0.579)	0.015 *
FM%	0.194 (0.028–0.36)	0.051
FMkg	0.131 (−0.021–0.284)	0.154
CentrFM	0.137 (−0.231–0.504)	0.939
FFM	−0.096 (−0.218–0.27)	0.078
FFMI	0.141 (−0.415–0.697)	0.083

* Statistically significant; ^#^ Non-standardized B coefficient.

**Table 3 medicina-57-00292-t003:** Multivariate linear regression analysis for CRP.

Risk Factors Examined	Multivariate R^2^	
	B (95%CI)	*p*
minSatO2	−0.017 (−0.215–0.182)	0.808
BMI	0.534 (−1.836–2.903)	0.525
FM%	1.306 (1.935–4.547)	0.290
FMI	−3.015 (−11.3–5.27)	0.331

## Data Availability

The data presented in this study are available on request from the corresponding author.

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
