# Peer review of "Does the Severity of Obstructive Sleep Apnea Have an Independent Impact on Systemic Inflammation?"

_medicina, 2021, doi:10.3390/medicina57030292_

Round 1
Reviewer 1 Report
A large sample and control group non obese are required.
Author Response
Dear Reviewer,
Thank you for your review and your observations and suggestions on improvement of the submitted manuscript. We have read and analyzed them carefully and considered the possibility to implement them in the manuscript. And, although we find them reasonable, it would not be possible to change the design of the research to such a scale in order to provide larger sample and control group of non-obese patients.
Even though we are aware of the limitations of this study, and we stated them in the manuscript, our intention was to analyze OSA patients according to severity of the disorder and the impact it has on systemic inflammation.
However, some changes to the manuscript have been made according the suggestions of the other reviewer and you may find them in the revised manuscript version, marked under “Track Changes” option.
We hope that you will take into consideration the revised version of the manuscript and that you will find it suitable enough for publication.
Thank you. Best regards,
The Authors
26 Feb 2021
Reviewer 2 Report
The aim of the study is to show whether there is association between sleep apnea syndrome severity and chronic inflammation or this previously established association is just caused by concomitant obesity. This question has been already many times analysed. However, the methods used in this study could justify publication of this paper but under the condition the Authors clearly state what is new in their study. The part “Limitation of the study” makes it questionable.
The title is misleading, as the results are negative; it should be corrected in order to mirror the findings of the study.
Methods. There should be more details about “sleep” study. Was it really sleep study or nocturnal respiratory polygraphy?
There is inconsistency, as once a group is described as AHI>15 (abstract) and once as AHI>15 (e.g. Tabl)
There are no “Graphs” in this study (as mentioned in the “Results”) – there is one Figure and the Tables. The technical quality of part “B” of the Figure should be improved
Discussion should be re-organized as it can not start with introductory information. The last sentence in the discussion is not clear: The authors have listed important methodological limitations of their own study indicating that one could not rely on the results! The limitations of the study should be corrected.
The language needs corrections. Some of mistakes are underlined .
Author Response
Dear Reviewer,
Thank you for your review and your observations and suggestions on improvement of the submitted manuscript. We have read and analyzed them carefully and considered the possibility to implement them in the manuscript. Here is the list of the changes that have been made:
- The title has been changed, in order to pose the question that was answered in the findings of the study.
- In ”Methods” section, ”polisomnography” has been changed into ”nocturnal respiratory polygraphy”.
- Inconsistencies regarding the description of AHI groups have been corrected.
- Graphs have been added (5). They have been omitted in error in the first submission.
- We have put in additional efforts to provide better technical quality of part “B” of the Figure, new version was used to replace the old one.
- Discussion has been re-organized and the last sentence in the discussion has been changed.
- The ”Limitations of the study” have been corrected.
- Some changed in the language have been noted and corrected.
You may find all the listed corrections in the revised manuscript version, marked under “Track Changes” option.
We hope that you will take into consideration the revised version of the manuscript and that you will find it suitable enough for publication.
Thank you. Best regards,
The Authors
26 Feb 2021
Round 2
Reviewer 1 Report
The manuscript has been significantly
improved and now warrants publication in Medicina.
Reviewer 2 Report
Dear Editor,
The Authors have corrected the manuscript which now is suitable for publication. After correction very small additional editorial errors appeared: in v.32 and 245: previous “for” was better than “in” appearing in new version; v.80 it should be “polygraphy”